# TiO_2_ Nanoparticle/Polyimide Nanocomposite for Ultrahigh-Temperature Energy Storage

**DOI:** 10.3390/nano12244458

**Published:** 2022-12-15

**Authors:** Xinrui Chen, Wenbo Zhu, Jianwen Chen, Qing Cao, Yingxi Chen, Dengyan Hu

**Affiliations:** 1School of Mechatronic Engineering and Automation, Foshan University, Foshan 528000, China; 2School of Electronic Information Engineering, Foshan University, Foshan 528000, China

**Keywords:** polyimide-based nanocomposite, ultrahigh-temperature energy storage, dielectric permittivity, dielectric loss, finite element simulation

## Abstract

With the development of electronic technology, there is an increasing demand for high-temperature dielectric energy storage devices based on polyimides for a wide range of applications. However, the current nanofillers/PI nanocomposites are used for energy harvesting at no more than 200 °C, which does not satisfy the applications in the oil and gas, aerospace, and power transmission industries that require an operating temperature of 250–300 °C. Therefore, we introduced a nanocomposite based on nonsolid TiO_2_ nanoparticles and polyimide (PI) with high energy storage performance at an ultrahigh temperature of 300 °C. The synergy of excellent dielectric properties and a high breakdown strength endowed the nanocomposite with a low loading content of 1 wt% and a high energy storage density of 5.09 J cm^−3^. Furthermore, we found that the nanocomposite could stably operate at 300 °C with an outstanding energy storage capability (2.20 J cm^−3^). Additionally, finite element simulations demonstrated that the partially hollow nanostructures of the nanofillers avoided the evolution of breakdown paths, which optimized the breakdown strength and energy storage performance of the related nanocomposites. This paper provides an avenue to broaden the application areas of PI-based nanocomposites as ultrahigh-temperature energy-storage devices.

## 1. Introduction

The rapid growth of electrical automotive, aerospace, and subsurface energy exploration requires a new generation of polymer-based capacitors with portability, high energy storage performance, and elevated operation temperatures [1,2,3,4]. In particular, parts of energy storage devices need to operate at a circumstantial temperature of or near 250–300 °C, such as the power and electronic systems used in the aforementioned industries [4,5]. Polyimides (PIs) are highly expected to be applied as key polymers in high-temperature energy harvesting because PIs have a higher glass transition temperature (Tg) of 360 °C than any other commonly-used dielectric polymers such as poly(ethylene imine) (less than 250 °C) [6,7]. Hence, PIs with high thermal stability are capable of overcoming the drawbacks of the current commercial biaxially-oriented polypropylene (BOPP) films, which can only operate at less than 105 °C [8]. In addition, PIs are linear dielectric polymers with excellent dielectric permittivity and low dielectric loss, and have the potential to yield a high energy storage density as well as a good charge–discharge efficiency [2].

High-temperature polymer dielectric materials with a relatively low dielectric permittivity limit their energy storage density. Due to their excellent dielectric properties [9,10,11] and their capability of withstanding high temperatures [12], low-dimensional titanium oxides are highly expected to be introduced into PI matrices to fabricate nanocomposites for improving their performance. Meanwhile, there is an interplay between the dielectric permittivity and breakdown strength, and this has an effect on the energy storage characteristics. For instance, Ai et al. studied diverse metal oxides nanodielectrics and found that a TiO_2_ nanofiller/PI nanocomposite with a low loading content (1 vol%) exhibited a higher dielectric permittivity (4.02) than other counterparts with HfO_2_ (3.76) and Al_2_O_3_ (3.58) at a higher loading [13]. However, the nanocomposite based on TiO_2_ yielded the lowest breakdown strength at 340 MV m^−1^, resulting in an insufficient energy-storage density of 1.48 J cm^−3^ at 150 °C [13]. Therefore, improving the energy storage properties of PI-based nanocomposites at low loading and high temperature is a great challenge.

Solid TiO_2_ nanoparticles are semiconductors with a high dielectric constant (≈40), good electrical conductivity, high stability, low cost, and non-toxicity, but they have a low specific surface area [14,15,16]. Nonsolid TiO_2_ nanoparticles are simple to prepare and are partially hollow structures with a large specific surface area, which can increase the interfacial polarization and be beneficial for improving the breakdown strength of the nanocomposites [17]. As reported in our previous article [17], the structural engineering of TiO_2_ nanoparticles is a promising way to enhance the breakdown strength as well as the energy storage density of nanocomposites, which may meet the urgent needs of high-temperature energy-storage devices. In this work, a nanocomposite based on nonsolid anatase TiO_2_ nanoparticles and PI was introduced to achieve high energy storage at an elevated temperature of 300 °C. The low loading (1 wt%) of partially hollow TiO_2_ nanofillers resulted in enhancing the dielectric permittivity and maintaining the breakdown strength of the nanocomposite compared with the pure PI matrix. The TiO_2_/PI nanocomposite showed a high energy-storage density of 5.09 and 2.20 J cm^−3^ at room temperature and a high temperature of 300 °C, respectively. The finite element simulation results demonstrated that the TiO_2_ and air interfaces of the nonsolid nanoparticles hindered the rapid evolution of the breakdown paths and lowered the electric potential in the nanocomposite, which played an important role in improving its breakdown strength as well as its energy-storage density. This work provides an effective strategy for the applications of PI-based nanocomposites in extremely high-temperature circumstances.

## 2. Materials and Methods

*Materials:* MXene (Ti_3_C_2_T*_x_*) nanosheets were provided by the State Key Laboratory of Optoelectronic Materials and Technologies, School of Electronics and Information Technology, Sun Yat-sen University, Guangzhou (China). Polyimide was provided by Dongguan Zhanyang Polymer Materials Co., Ltd. (Dongguan, China). N, N-Dimethylformamide (DMF) (99.5%) and Hydrogen peroxide solution (30 wt%) were provided by Shanghai Aladdin Biochemical Technology Co., Ltd. (Shanghai, China).

*Fabrication of TiO_2_/PI nanocomposite:* The process schematics of the prepared TiO_2_/PI nanocomposite are shown in Figure 1a. Polyimide is a linear dielectric material, and its glass transition temperature (Tg) is about 310 °C, which is consistent with the DSC curve in Appendix A. TiO_2_ nanoparticles were prepared by the H_2_O_2_-assisted oxidation of MXene (Ti_3_C_2_T*_x_*) nanosheets at room temperature (RT), as reported in our recent article [17]. Subsequently, TiO_2_/PI nanocomposites with mass fractions of 1, 2, and 5 wt% were fabricated by solution blending and spin coating, respectively. 

For the preparation of TiO_2_ nanoparticles/PI nanocomposites with different loading of nanofillers, firstly three same PI/N, N-Dimethylformamide (DMF) solutions were obtained by adding 0.04 g PI powders into three beakers with 8 mL DMF solvent and stirring the mixtures with a magnetic stirrer for the uniform solution. Next, 0.01, 0.02, and 0.05 g TiO_2_ nanoparticles were added to the PI/DMF solutions, respectively, then dispersed by a 4 h stirring process and 3 min ultrasonication. Afterwards, 0.96 g PI powders were dispersed into the three mixtures by the same stirring and ultrasonication processes, respectively, to obtain three TiO_2_/PI/DMF dispersions with different contents of TiO_2_. Finally, these dispersions were spin-coated onto ITO glass substrates at the speed of 500 rad min^−1^ for 5 s and 3500 rad min^−1^ for 25 s (KW-4A, SETCAS Electronics Co., Ltd., Beijing, China), respectively, and dried in a vacuum oven at 150 °C for 2h to obtain TiO_2_ nanoparticles/PI nanocomposites with different loading contents of 1, 2, and 5 wt%.

*Characterization of TiO_2_/PI nanocomposite:* The morphological and structural characterization of Ti_3_C_2_T*_x_* nanosheets and TiO_2_ nanoparticles were performed by scanning electron microscopy (SEM, Zeiss Sigma 300), transmission electron microscopy (TEM, FEI Tecnai F20), and corresponding SAED. The lattice structure of the materials was analyzed by X-ray diffraction (XRD, X’Pert PRO MPD) and Raman spectroscopy (Raman, HR800, Horiba JY). Meanwhile, X-ray photoelectron spectroscopy (XPS, Thermo Kalpha, Guangzhou, China) was used to characterize the chemical valence state of the materials. The thermal stability of the nanocomposites was tested by the Differential scanning calorimeter (DSC, TA, NETZSCH) and Thermogravimetric analysis (TGA). The Tg was obtained with a heating rate of 10 °C min^−1^ from RT to 500 °C in a nitrogen atmosphere. TGA was implemented using a thermal analyzer (RIGAKU, TG-DTA8122, Thermo plus EVO2) to test the thermal stability with a heating rate of 20 °C min^−1^ from RT to 800 °C in an atmosphere of nitrogen. The dielectric properties of TiO_2_ nanoparticles/PI nanocomposites versus frequency (1 to 10^3^ kHz) and temperature (RT to 300 °C) were investigated using a precision impedance analyzer (E4980A, Agilent). A high-vacuum multi-target magnetron sputtering coating system (JCP350PM, Beijing Technol Science Co., Ltd., Beijing, China) was used for the deposition of circular aluminum electrode arrays (300 nm in thickness and 1 mm in diameter for each electrode) at a rate of 10 nm min^−1^. The schematic diagram of the measurement device for measuring the breakdown strength of nanocomposites is shown in Figure 4a. Circular Al electrodes with a diameter of 1 mm and a thickness of 300 nm were deposited on the film to apply a voltage through a probe, while the bottom ITO glass served as another electrode. The breakdown strengths of the nanocomposites with different TiO_2_ contents were analyzed according to the Weibull distribution function: $P(E)=1-\exp[-{\left(E/E_{b}\right)}^{\beta }]$ [18], where *P*(*E*) is the cumulative probability of electric failure, *E* is the measured breakdown strength, *E*_b_ is the field strength at a cumulative failure probability of 63.2%, and *β* is the shape parameter. 

*Finite element simulation:* To further analyze the effect of the structure of TiO_2_ nanoparticles on the breakdown strength of PI-based nanocomposites, finite element simulations were carried out. In the simulation system, TiO_2_ nanoparticles were randomly distributed in the PI matrix. The constructed composite model is discretized into a two-dimensional array of 800 × 1000 grid points before implementation. Additionally, the dielectric permittivity of PI and TiO_2_ nanoparticles was set based on the actual values measured here.

## 3. Results and Discussion

As shown in Figure 1a, the Ti_3_C_2_T*_x_* nanosheets were oxidized with the assistance of H_2_O_2_ at RT to prepare the TiO_2_ nanoparticles, which were mixed with PI/DMF solution. Subsequently, the dispersion was spin-coated onto the ITO substrate to obtain a TiO_2_/PI nanocomposite after the evaporation of DMF. The detailed preparation process in Figure 1a is described in *Experimental Section*. The scanning electron microscopy (SEM) of Figure 1b shows that the lateral size of Ti_3_C_2_T*_x_* is about 20 μm and is lamellar in the low-magnification TEM image (Figure 1c). It is observed in Figure 1d that the d-spacing of 0.35 nm is corresponding to the (006) plane of the Ti_3_C_2_T*_x_* Mxene, which is consistent with the results reported in previous literature [19]. The selected-area electron diffraction (SAED) pattern of Figure 1e shows the hexagonal symmetry of the Ti_3_C_2_T*_x_* nanosheets and further demonstrates their high quality. After the oxidation process by H_2_O_2_ solution at room temperature, the scanning electron microscope (SEM) image in Figure 1f shows that the original two-dimensional morphology is changed to nanoparticles with a diameter of hundreds of nanometers. It indicates that the Ti_3_C_2_T*_x_* nanosheets were wrinkled during the chemical reaction, and that the nanoparticles may be nonsolid. A similar phenomenon was also reported in other articles [20,21]. Moreover, the low-magnification SEM image shows a good dispersion of TiO_2_ nanoparticles (Appendix A). In addition, the SEM images as shown in Appendix A indicate that the TiO_2_ nanoparticles are uniformly distributed in the PI matrix. This is further confirmed by the uniform distribution of the Ti elements shown in EDS mapping (Appendix A). The high-magnification TEM image (Figure 1g) demonstrates that the TiO_2_ *d*-spacing in the (101) plane is 0.35 nm, and the SAED (Figure 1h) pattern shows distinct diffraction rings which are consistent with the anatase phase of TiO_2_. Therefore, it is facile to produce partially hollow TiO_2_ nanoparticles with high quality at ambient temperature via H_2_O_2_-assisted oxidation. Finally, the cross-sectional SEM image in Figure 1i clarifies that the thickness of the TiO_2_/PI nanocomposite is ≈2 μm, which is used to calculate the breakdown strength afterward. 

X-ray diffraction (XRD) is further used to characterize the lattice structure of the TiO_2_ nanoparticles. Figure 2a exhibits a (002) peak for Ti_3_C_2_T*_x_* nanosheets exfoliated from raw materials [22], but the (006) peak is too weak to be detected by XRD [23]. In comparison, (101), (004), (200), (105), (211), and (204) with high intensity are collected, while some weak peaks belonging to other unindexed TiO_2_ are also observed [24]. XRD analysis indicated that the major anatase phase of the TiO_2_ produced according to its standard PDF card of TiO_2_ (JCPDS, 21-1272), which is in agreement with the HRTEM image (Figure 1g). 

Furthermore, Raman spectroscopy is employed to confirm the phase of the TiO_2_ nanoparticles. As shown in Figure 2b, the A_1g_ peak (205 cm^−1^) of the Ti_3_C_2_T*_x_* nanosheets is observed, which is reported by other previous articles [25]. After the oxidation of the Ti_3_C_2_T*_x_* nanosheets, four characteristic peaks are located at 144, 415, 515, and 650 cm^−1^ corresponding to the *E*_g_, *B*_1g_, *A*_1g_, and another *E*_g_ modes of the typical anatase TiO_2_, respectively [26,27]. In addition, it is essential to characterize the chemical composition of TiO_2_ nanoparticles by X-ray photoelectron spectroscopy (XPS). The XPS spectrum of the C 1s core levels of Ti_3_C_2_T*_x_* nanosheets indicates the existence of C-Ti (282.0 eV), C-C (284.8 eV), C=O (286.3 eV), and O-C=O, C-F (288.9 eV) bonds, in which F element is induced during the preparation process of MXene [28]. Ti 2p element is located by Ti-C (454.3 and 461.0 eV corresponding to Ti 2p_3/2_ and Ti 2p_1/2_, respectively) and Ti-O bonds (458.8 and 464.7 eV corresponding to Ti 2p_3/2_ and Ti 2p_1/2_, respectively), as shown in Figure 2d. In contrast, Ti-C bonds cannot be detected by XPS, while the intensity of Ti-O signals increases dramatically, as plotted in Figure 2e,f. Ti 2p component centered at 458.8 eV is associated with Ti ions in a +4 valence state [29]. The Ti-O signals are detected by XPS (3) after mixing and the Ti 2p component is centered at 458.8 eV, associated with +4 valence Ti ions. It is indicated that it does not have much effect on the TiO_2_ nanoparticles after mixing. Consequently, these results demonstrate that the Ti_3_C_2_T*_x_* nanosheets are completely oxidized and that the major product is anatase TiO_2_ nanoparticles. 

Dielectric properties of the TiO_2_/PI nanocomposite play a vital role in its energy storage performance. Figure 3a shows the dielectric permittivity of the PI matrix and TiO_2_/PI nanocomposites at different loading contents. Pure PI has a dielectric permittivity of ≈3.28 at 1 kHz, which is comparable to that reported by previous articles [30]. The introduction of 1 wt% TiO_2_ nanofillers improves this parameter of the related nanocomposites to ≈5.91. Moreover, the dielectric permittivity is positively correlated to the loading content and increases to ≈7.53 of the TiO_2_/PI nanocomposite at 5 wt% loadings. The improved dielectric permittivity is ascribed to more interfaces created by the non-solid TiO_2_ nanoparticles as well as their uniform distribution in the polymer matrix. Note that the dielectric permittivity of all nanocomposites and PI matrix almost keep stable with the increasing frequency from 1 to 10^3^ kHz. On the other hand, the dielectric loss of the nanocomposites with the addition of 1 wt% loading content of TiO_2_ nanoparticles is slightly increased and less than 0.01 at the frequency of 1 kHz, which is considered to be beneficial for the development of energy storage devices.

To investigate the effect of temperature on the dielectric properties of TiO_2_/PI nanocomposites, the dielectric permittivity and loss were measured over a wide temperature range (RT to 300 °C). It is found that the dielectric permittivity of the nanocomposites and PI matrix show a similar decreasing trend with the raising temperature. It is noteworthy that the nanocomposites drop before pure PI, the addition of the filler phase accelerates the thermal expansion of the polymer, leading to the breakage of the polymer chains, and reducing the energy required for the molecules [31]. Furthermore, the introduction of TiO_2_ nanoparticles into the PI matrix increases the porosity of the nanocomposites, which also leads to a decrease in the dielectric permittivity [32]. The dielectric permittivity of the nanocomposite at a low loading of 1 wt% is reduced by 20%, but still more than 5 at 300 °C. It is higher than that of the 1 vol% content TiO_2_ nanofibers/PI nanocomposite (≈3.25) [13]. Meanwhile, the increasing temperature leads to the increase of the dielectric loss with fluctuations at elevated section, which is less than 0.02 for the nanocomposites with 1 and 2 wt% TiO_2_ nanofillers at 300 °C. It can be attributed to the accelerated thermal movement of molecules increasing at high temperatures, leading to the conductivity loss increasing sharply [33]. Additionally, as shown in Appendix A, the dielectric permittivity of the solid TiO_2_/PI nanocomposites is lower than that of the non-solid ones. In sum, the TiO_2_/PI nanocomposites at low loading show outstanding dielectric properties at elevated temperatures of 300 °C, which indicates their great potential for the fabrication of high-temperature energy storage devices.

Breakdown strength is another factor influencing the dielectric energy storage performance. The exact method of operation of the measurement device in Figure 4a has been pointed out in *Experimental Section*. As shown in Figure 4b, a decreasing tendency of the breakdown strength of the nanocomposites with the increase in temperature because of the accumulation of Joule heat at elevated temperature and high electric field [18]. In comparison to pure PI, the TiO_2_/PI nanocomposite at 1 wt% loading shows comparable breakdown features at the same temperatures (specifically 441 and 290 MV m^−1^ at RT and 300 °C, respectively). On the contrary, a larger content of TiO_2_ nanoparticles triggers the dramatic reduction of the breakdown strength of the related nanocomposites, because more nanofillers may result in aggregation and introduce more structural imperfections [11]. Consequently, low loading of nanodielectrics facilitates the maintenance of the breakdown strength as well as the thermal stability of the TiO_2_/PI nanocomposites at an elevated temperature of 300 °C.

In order to reveal the application potential of the TiO_2_/PI nanocomposites in high-temperature energy harvesting, the energy storage density is essential to be calculated. Considering the linear dielectric property of PI [2], the energy storage density of the related nanocomposites can be calculated by the following equation [34]:(1)$$U_{e}=\frac{1}{2}\varepsilon_{0}\varepsilon_{r}E_{b}^{2}$$
where *U*_e_ is the energy storage density, *ε*_0_ is the dielectric permittivity of vacuum, *ε*_r_, and *E*_b_ represent the relative permittivity and breakdown strength of the TiO_2_/PI nanocomposites. Accordingly, as shown in Figure 4c, the energy storage density of pure PI and TiO_2_/PI nanocomposites decreases with increasing temperature and filler contents, which is due to the substantial decrease in breakdown strength. The nanocomposite with 1 wt% loading content exhibits the highest energy storage density of 5.09 J cm^−3^ at RT. Its performance (3.31 J cm^−3^ at 200 °C) is more than twice that of another TiO_2_/PI counterpart (1.48 J cm^−3^ at 150 °C). In addition, the energy storage density is maintained at 2.20 J cm^−3^ at a high temperature of 300 °C. The TGA curves in Appendix A show that the pure PI and the TiO_2_/PI nanocomposite with 1 wt% content have no significant weight loss at less than 485 °C, indicating that the incorporation of TiO_2_ nanoparticles did not reduce the high-temperature resistance of pure PI, reflecting their excellent thermal stability. The DSC curves (Appendix A) show that the pure PI is heat-absorbent at 305.14 °C, which corresponds to the Tg of the pure PI, and the Tg of the TiO_2_/PI nanocomposite is about the same as the PI, further proving the thermal stability. Figure 4d summarizes the energy storage density of PI-based composites at RT and high temperatures [13,35,36,37,38,39,40,41]. On the one hand, the energy storage density (5.09 J cm^−3^) of the TiO_2_/PI nanocomposite with 1 wt% nanofillers at ambient temperature, is 500% higher than that of the BaTiO_3_ nanowires/PI nanocomposite with a volume fraction of 2 vol% [37]. Furthermore, the TiO_2_ nanoparticles/PI nanocomposite has a higher energy storage density of 2.20 J cm^−3^ at an ultrahigh temperature of 300 °C, while the other PI-based nanocomposites operate at no more than 200 °C. In summary, TiO_2_/PI nanocomposites with the combination of high-energy storage density and elevated operating temperature of 300 °C are promising for high-temperature energy storage in electric devices.

**Figure 4 nanomaterials-12-04458-f004:**
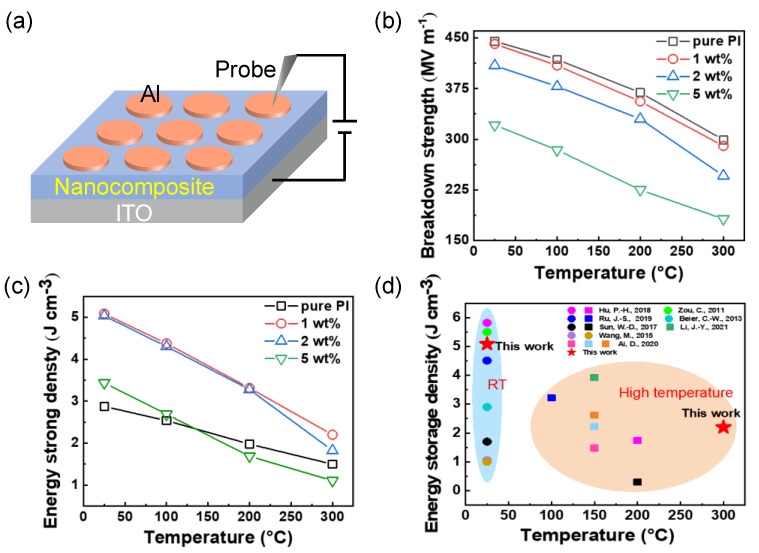
Energy storage performance of TiO_2_/PI nanocomposite. (**a**) Schematic setup for the measurement of breakdown strength. (**b**) Breakdown strength and (**c**) Energy storage density of the TiO_2_/PI nanocomposites with 1, 2, and 5 wt% loading and pure PI at various temperatures. (**d**) Comparison of the energy storage density of the PI-based composites at RT and high temperature [13,35,36,37,38,39,40,41].

Based on the structural characterization of TiO_2_ nanoparticles and the thickness of the TiO_2_/PI nanocomposite, a two-dimensional model was designed for finite element analysis. The relative permittivity of PI was set to 3.5, the ratio of the relative dielectric permittivity of the inner and outer parts of the nonsolid TiO_2_ nanoparticles was set to 1:100, and the permittivity of the solid TiO_2_ nanoparticles was set to 100. For linear dielectrics, we normalized the control equation for better numerical simulation. The variables and analytic functions in the model are set according to the following two equations. The dimensionless form of which can be written as [42]:(2)$$\bar{\nabla }\;\cdot \;\left[\frac{1}{f(s)+\eta }\bar{\nabla \varphi }\right]=0$$
(3)$$\frac{\partial s}{\partial \bar{t}}=-\frac{f^{\prime }(s)}{2{\left[f(s)+\eta \right]}^{2}}\bar{\nabla \varphi }\;\cdot \;\bar{\nabla \varphi }+f^{\prime }(s)+\frac{1}{2}\;{\bar{\nabla }}^{2}s=0$$
where *s* is a scalar phase field, *f* (*s*) = 4*s*^3^ − 3*s*^4^ is an interpolation function with values between 0 and 1 [43], *η* is a small enough number, *φ* and *t* represent the electric potential and time, respectively, and the corresponding quantities are symbolized with over-bars, respectively. 

Finite element simulation was implemented to study the positive impact of the nonsolid nanoparticles on the breakdown strength as well as the energy storage of the related nanocomposites. In the 2D model, the PI matrix was set to a two-dimensional array of 800 × 1000 grid points. Subsequently, the solid and nonsolid TiO_2_ nanoparticles were randomly created therein. Here, two different setups, namely nanocomposites with solid and hollow nanoparticles at the same loading content were used to simulate the breakdown process with time, respectively. Accordingly, it is reasonable that the number of nonsolid nanoparticles in the related nanocomposite is more than that in the solid case. 

Figure 5a,b show the time-dependent breakdown evolution of the TiO_2_/PI nanocomposite with solid and hollow nanofillers, respectively. The breakdown paths are indicated by the white trees, which extend from the equipotential lines with high electric potential energy to the low-energy ones. In the solid case, the breakdown starts at the top of the nanocomposite when the electric field reaches a critical value, then the electric trees grow and directly penetrate through the nanoparticles. During this process, the electric field is redistributed, and the strongest is at the tip of the breakdown path [44]. On the contrary, Figure 5b shows the lower electric potential between the hollow nanoparticles. The TiO_2_ and air interfaces facilitate the extension of the electric trees and produce more branches, which delay the spread of the breakdown paths, and enhance the breakdown strength. It is also shown that the hollow-structured TiO_2_ nanoparticles are more beneficial to enhance the breakdown strength of the composites compared to solid TiO_2_. 

Simultaneously, the polarization diagram in Appendix A demonstrates that the nanocomposite with hollow-structured TiO_2_ has a larger electric displacement under the same electric field. Because the hollow structure has more interfaces, the accelerated rate of internal charge movement leads to greater interfacial polarization. It is shown that the hollow structure of TiO_2_ nanoparticles enhanced the interfacial polarization of the nanocomposite, which increased the breakdown strength. The results of the simulations are in agreement with the experimental ones. Additionally, Figure 5c shows the change of nominal electric field with the charge density in the nanocomposites with solid and hollow nanofillers, which is corresponding to Figure 5a,b. It can be seen that the maximum breakdown strength of nanocomposites with nonsolid nanostructure is larger than that based on solid ones. Similarly, the maximum polarizability (Appendix A) of the nanocomposite with hollow nanofillers is larger than that based on solid ones. From the above, it can be seen that nonsolid nanoparticles are more beneficial in enhancing the breakdown strength than solid nanoparticles. Therefore, the improvement of the breakdown strength and energy storage is beneficial from the nonsolid nanoparticles, which create more interfaces to prevent the electric trees from rapid growth and reduce the spatial electric potential in the nanocomposite.

## 4. Conclusions

We have fabricated a nanocomposite based on nonsolid TiO_2_ nanoparticles and PI for ultrahigh-temperature energy storage. The dielectric permittivity is ≈5.91 of the TiO_2_/PI nanocomposite and exhibits excellent frequency stability. In addition, the nanocomposite shows a breakdown strength of 441 MV m^−1^ and an energy storage density of 5.09 J cm^−3^ at RT. Moreover, the operating temperature of the PI-based nanocomposite has been raised to 300 °C with an energy storage density of 2.20 J cm^−3^. It is shown that the non-solid structure of TiO_2_ nanoparticles facilitates energy storage at high temperatures. Additionally, finite element simulation results illustrated that the partially hollow nanostructure of the TiO_2_ nanodielectrics increases the interfacial polarization of the nanocomposites, which contributes to optimizing the breakdown strength and improving their energy storage performance. This work presents the use of the hollow structure of TiO_2_ nanoparticles to improve the energy storage properties of nanocomposites at high temperatures under low loading, paving the way to broaden the application of PI-based energy storage devices to the aerospace industry, underground energy exploration, and electricity electronics in the ultrahigh-temperature circumstances of 300 °C.

## Figures and Tables

**Figure 1 nanomaterials-12-04458-f001:**
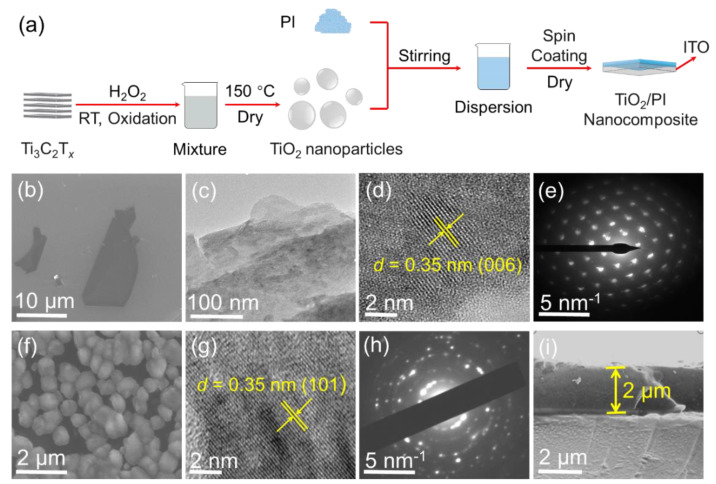
Preparation of TiO_2_/PI nanocomposites. (**a**) Process schematics for the preparation of TiO_2_/PI nanocomposites. (**b**) SEM image, (**c**) low-magnification TEM image, (**d**) high-magnification TEM image, and (**e**) SAED pattern of Ti_3_C_2_T*_x_*. (**f**) SEM image, (**g**) high-magnification TEM image, and (**h**) SAED pattern of TiO_2_ nanoparticles. (**i**) Cross-sectional SEM image of TiO_2_/PI nanocomposite.

**Figure 2 nanomaterials-12-04458-f002:**
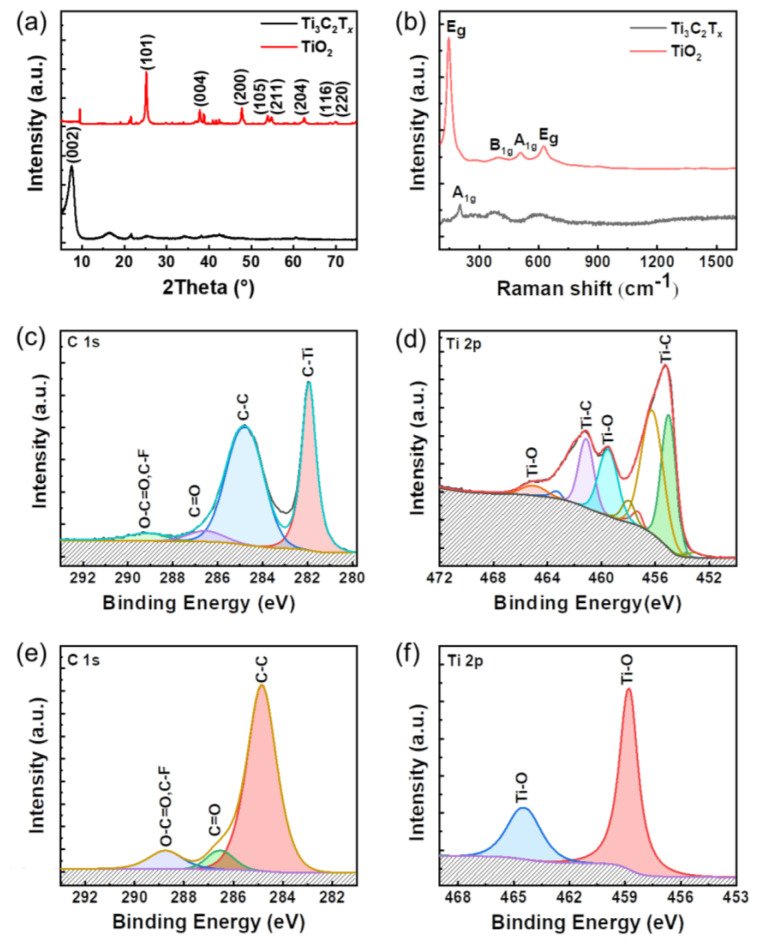
Characterization of Ti_3_C_2_T_x_ nanosheets and TiO_2_ nanoparticles. (**a**) XRD and (**b**) Raman spectra of Ti_3_C_2_T_x_ nanosheets and TiO_2_ nanoparticles. XPS spectra of (**c**) C 1s and (**d**) Ti 2p components of Ti_3_C_2_T*_x_* nanosheets. XPS spectra of (**e**) C 1s and (**f**) Ti 2p components of TiO_2_ nanoparticles.

**Figure 3 nanomaterials-12-04458-f003:**
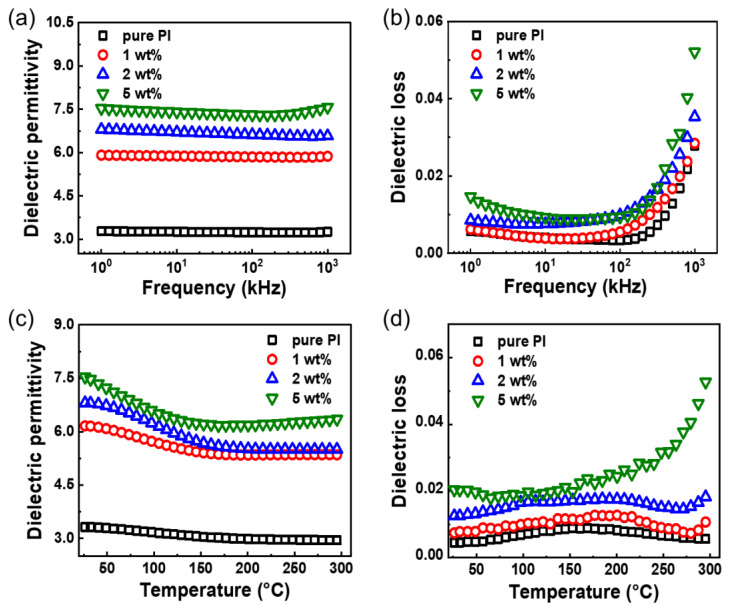
Dielectric properties of TiO_2_/PI nanocomposite. Frequency-dependent (**a**) dielectric permittivity and (**b**) dielectric loss of pure PI and TiO_2_/PI nanocomposites with various loading contents of 1, 2, and 5 wt% at RT. Temperature-dependent (**c**) dielectric permittivity and (**d**) dielectric loss of pure PI and TiO_2_/PI nanocomposites with various loading contents of 1, 2, and 5 wt% at 1 kHz.

**Figure 5 nanomaterials-12-04458-f005:**
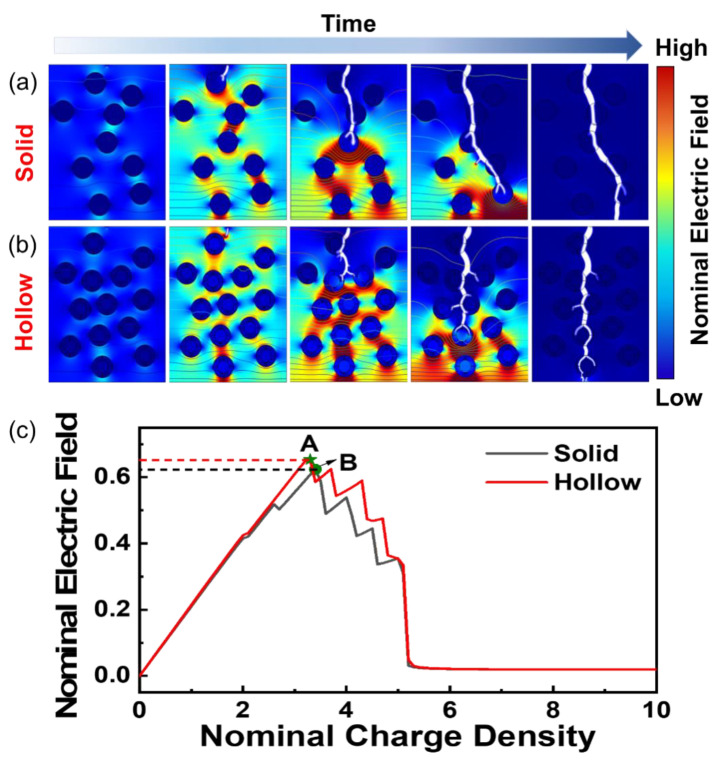
Finite element analysis of energy storage performance of TiO_2_/PI nanocomposites. Finite element simulation of breakdown evolution of TiO_2_/PI nanocomposites with (**a**) solid and (**b**) hollow nanoparticles. (**c**) Relationship between nominal electric field and charge density for PI-based nanocomposites with solid (grey curve) and hollow (red curve) nanoparticles.

## Data Availability

The data presented in this study are available from the corresponding author on request.

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
