# Peer review of "TiO2 Nanoparticle/Polyimide Nanocomposite for Ultrahigh-Temperature Energy Storage"

_nanomaterials, 2022, doi:10.3390/nano12244458_

Round 1
Reviewer 1 Report
My suggestions are:
1. The sentences in lines from 54 to 62 are some conclusions. It should not be a part of introduction.
2. The aim of paper is not clear. What does it mean to introduce your material?
3. It is not possible measure chemical composition using XRD. You measured mineral composition.
4. The part from 107 to 110 should be omitted. Fig. 1a must be cited in experimental part.
5. Conclusion must be improved.
6. For me, there is missing DSC analysis of samples.
7. The format of unit in Fig. 4d must be corrected.
8. Fig. 4a should be cited in experimental part.
9. Experimental part must be improved.
Round 2
Reviewer 1 Report
Authors improved the manuscript. Nevertheless, several corrections must be make.
Why did authors provide DSC measurements only up to 500 °C and TGA up to 800 °C? Authors should connect DSC and TGA measurements. I recommend to provide DSC up to 800 °C.
Formal corrections:
Line 97: There should be NETZSCH.
Fig. S4: The unit of temperature is missing.
Round 3
Reviewer 2 Report
The authors brought additional improvements to the manuscript, and some of (not all) the queries were answered.
In its current form, the manuscript comes close to a publishable one.
I still believe the structure of the Pi used is necessary and further structure-properties correlations are in order.
More important, I underline the necessity to perform a thorough English check and revision.
